# Approximate maximum entropy principles via Goemans-Williamson with applications to provable variational methods

**Yuanzhi Li**
Department of Computer Science
Princeton University
Princeton, NJ, 08450
yuanzhil@cs.princeton.edu

**Andrej Risteski**
Department of Computer Science
Princeton University
Princeton, NJ, 08450
risteski@cs.princeton.edu

## Abstract

The well known maximum-entropy principle due to Jaynes, which states that given mean parameters, the maximum entropy distribution matching them is in an exponential family has been very popular in machine learning due to its "Occam's razor" interpretation. Unfortunately, calculating the potentials in the maximum-entropy distribution is intractable [BGS14]. We provide *computationally efficient* versions of this principle when the mean parameters are pairwise moments: we design distributions that approximately match given pairwise moments, while having entropy which is comparable to the maximum entropy distribution matching those moments.

We additionally provide surprising applications of the approximate maximum entropy principle to designing provable variational methods for partition function calculations for Ising models *without any assumptions* on the potentials of the model. More precisely, we show that we can get approximation guarantees for the log-partition function comparable to those in the low-temperature limit, which is the setting of *optimization* of quadratic forms over the hypercube. ([AN06])

## 1 Introduction

**Maximum entropy principle** The maximum entropy principle [Jay57] states that given *mean parameters*, i.e. $\mathbb{E}_\mu[\phi_t(\mathbf{x})]$ for a family of functionals $\phi_t(\mathbf{x}), t \in [1, T]$, where $\mu$ is distribution over the hypercube $\{-1, 1\}^n$, the entropy-maximizing distribution $\mu$ is an exponential family distribution, i.e. $\mu(\mathbf{x}) \propto \exp(\sum_{t=1}^T J_t \phi_t(\mathbf{x}))$ for some *potentials* $J_t, t \in [1, T]$. [1] This principle has been one of the reasons for the popularity of graphical models in machine learning: the "maximum entropy" assumption is interpreted as "minimal assumptions" on the distribution other than what is known about it.

However, this principle is problematic from a computational point of view. Due to results of [BGS14, SV14], the potentials $J_t$ of the Ising model, in many cases, are impossible to estimate well in polynomial time, unless NP = RP – so merely getting the description of the maximum entropy distribution is already hard. Moreover, in order to extract useful information about this distribution, usually we would also like to at least be able to sample efficiently from this distribution – which is typically NP-hard or even #P-hard.

In this paper we address this problem in certain cases. We provide a "bi-criteria" approximation for the special case where the functionals $\phi_t(\mathbf{x})$ are $\phi_{i,j}(\mathbf{x}) = \mathbf{x}_i\mathbf{x}_j$, i.e. pairwise moments: we produce a efficiently sampleable distribution over the hypercube which matches these moments up to multiplicative constant factors, and has entropy at most a constant factor smaller from from the entropy of the maximum entropy distribution. [2]

Furthermore, the distribution which achieves this is very natural: the sign of a multivariate normal variable. This provides theoretical explanation for the phenomenon observed by the computational neuroscience community [BB07] that this distribution (there named *dichotomized Gaussian* there) has near-maximum entropy.

**Variational methods** The above results also allow us to get results for a seemingly unrelated problem – approximating the *partition function* $\mathcal{Z} = \sum_{\mathbf{x} \in \{-1,1\}^n} \exp(\sum_{t=1}^{T} J_t\phi_t(\mathbf{x}))$ of a member of an exponential family. The reason this task is important is that it is tied to calculating marginals.

One of the ways this task is solved is variational methods: namely, expressing $\log \mathcal{Z}$ as an optimization problem. While there is a plethora of work on variational methods, of many flavors (mean field, Bethe/Kikuchi relaxations, TRBP, etc. for a survey, see [WJ08]), they typically come either with no guarantees, or with guarantees in very constrained cases (e.g. loopless graphs; graphs with large girth, etc. [WJW03, WJW05]). While this is a rich area of research, the following extremely basic research question has not been answered:

*What is the best approximation guarantee on the partition function in the worst case (with no additional assumptions on the potentials)?*

In the low-temperature limit, i.e. when $|J_t| \to \infty$, $\log \mathcal{Z} \to \max_{\mathbf{x} \in \{-1,1\}^n} \sum_{t=1}^{T} J_t\phi_t(\mathbf{x})$ - i.e. the question reduces to purely optimization. In this regime, this question has very satisfying answers for many families $\phi_t(\mathbf{x})$. One classical example is when the functionals are $\phi_{i,j}(\mathbf{x}) = \mathbf{x}_i\mathbf{x}_j$. In the graphical model community, these are known as Ising models, and in the optimization community this is the problem of optimizing quadratic forms and has been studied by [CW04, AN06, AMMN06].

In the optimization version, the previous papers showed that in the worst case, one can get $O(\log n)$ factor multiplicative factor approximation of it, and that unless P = NP, one cannot get better than constant factor approximations of it.

In the finite-temperature version, it is known that it is NP-hard to achieve a $1 + \epsilon$ factor approximation to the partition function (i.e. construct a FPRAS) [SS12], but nothing is known about coarser approximations. We prove in this paper, informally, that one can get comparable multiplicative guarantees on the *log-partition function* in the finite temperature case as well – using the tools and insights we develop on the maximum entropy principles.

Our methods are extremely generic, and likely to apply to many other exponential families, where algorithms based on linear/semidefinite programming relaxations are known to give good guarantees in the optimization regime.

## 2 Statements of results and prior work

**Approximate maximum entropy** The main theorem in this section is the following one.

**Theorem 2.1.** *For any covariance matrix $\Sigma$ of a centered distribution $\mu : \{-1, 1\}^n \to \mathbb{R}$, i.e. $\mathbb{E}_\mu[\mathbf{x}_i\mathbf{x}_j] = \Sigma_{i,j}$, $\mathbb{E}_\mu[\mathbf{x}_i] = 0$, there is an efficiently sampleable distribution $\tilde{\mu}$, which can be sampled as $sign(g)$, where $g \sim \mathcal{N}(0, \Sigma + \beta I)$ and satisfies $\dfrac{\mathcal{G}}{1 + \beta}\Sigma_{i,j} \leq E_{\tilde{\mu}}[X_iX_j] \leq \dfrac{1}{1 + \beta}\Sigma_{i,j}$ and has entropy $H(\tilde{\mu}) \geq \dfrac{n}{25}\dfrac{(3^{1/4}\sqrt{\beta}-1)^2}{\sqrt{3}\beta}$, for any $\beta \geq \dfrac{1}{3^{1/2}}$.*

There are two prior works on computational issues relating to maximum entropy principles, both proving hardness results.

[BGS14] considers the "hard-core" model where the functionals $\phi_t$ are such that the distribution $\mu(\mathbf{x})$ puts zero mass on configurations $\mathbf{x}$ which are not independent sets with respect to some graph $G$.

They show that unless NP = RP, there is no FPRAS for calculating the potentials $J_t$, given the mean parameters $\mathbb{E}_\mu[\phi_t(\mathbf{x})]$.

[SV14] prove an equivalence between calculating the mean parameters and calculating partition functions. More precisely, they show that given an oracle that can calculate the mean parameters up to a $(1 + \epsilon)$ multiplicative factor in time $O(\text{poly}(1/\epsilon))$, one can calculate the partition function of the same exponential family up to $(1 + O(\text{poly}(\epsilon)))$ multiplicative factor, in time $O(\text{poly}(1/\epsilon))$. Note, the $\epsilon$ in this work potentially needs to be polynomially small in $n$ (i.e. an oracle that can calculate the mean parameters to a fixed multiplicative constant cannot be used.)

Both results prove hardness for *fine-grained* approximations to the maximum entropy principle, and ask for outputting approximations to the mean parameters. Our result circumvents these hardness results by providing a distribution which is not in the maximum-entropy exponential family, and is allowed to only approximately match the moments as well. To the best of our knowledge, such an approximation, while very natural, has not been considered in the literature.

**Provable variational methods** The main theorems in this section will concern the approximation factor that can be achieved by degree-2 pseudo-moment relaxations of the standard variational principle due to Gibbs. ([Ell12]) As outlined before, we will be concerned with a particularly popular exponential family: Ising models. We will prove the following three results:

**Theorem 2.2** (Ferromagnetic Ising, informal)**.** *There is a convex programming relaxation based on degree-2 pseudo-moments that calculates up to multiplicative approximation factor 50 the value of* $\log \mathcal{Z}$ *where $\mathcal{Z}$ is the partition function of the exponential distribution* $\mu(\mathbf{x}) \propto \exp(\sum_{i,j} J_{i,j} \mathbf{x}_i \mathbf{x}_j)$ *for*

$J_{i,j} > 0$.

**Theorem 2.3** (Ising model, informal)**.** *There is a convex programming relaxation based on degree-2 pseudo-moments that calculates up to multiplicative approximation factor $O(\log n)$ the value of* $\log \mathcal{Z}$ *where $\mathcal{Z}$ is the partition function of the exponential distribution* $\mu(\mathbf{x}) \propto \exp(\sum_{i,j} J_{i,j} \mathbf{x}_i \mathbf{x}_j)$.

**Theorem 2.4** (Ising model, informal)**.** *There is a convex programming relaxation based on degree-2 pseudo-moments that calculates up to multiplicative approximation factor $O(\log \chi(G))$ the value of* $\log \mathcal{Z}$ *where $\mathcal{Z}$ is the partition function of the exponential distribution* $\mu(\mathbf{x}) \propto \exp(\sum_{i,j \in E(G)} J_{i,j} \mathbf{x}_i \mathbf{x}_j)$

*where $G = (V(G), E(G))$ is a graph with chromatic number $\chi(G)$.* [3]

While a lot of work is done on variational methods in general (see the survey by [WJ08] for a detailed overview), to the best of our knowledge nothing is known about the worst-case guarantee that we are interested in here. Moreover, other than a recent paper by [Ris16], no other work has provided provable bounds for variational methods that proceed via a convex relaxation and a rounding thereof.[4]

[Ris16] provides guarantees in the case of Ising models that are also based on pseudo-moment relaxations of the variational principle, albeit only in the special case when the graph is "dense" in a suitably defined sense. [5] The results there are very specific to the density assumption and can not be adapted to our worst-case setting.

Finally, we mention that in the special case of the ferromagnetic Ising models, an algorithm based on MCMC was provided by [JS93], which can give an approximation factor of $(1 + \epsilon)$ to the partition function and runs in time $O(n^{11}\text{poly}(1/\epsilon))$. In spite of this, the focus of this part of our paper is to provide understanding of variational methods in certain cases, as they continue to be popular in practice for their faster running time compared to MCMC-based methods but are theoretically much more poorly studied.

# 3 Approximate maximum entropy principles

Let us recall what the problem we want to solve:

**Approximate maximum entropy principles** We are given a positive-semidefinite matrix $\Sigma \in \mathbb{R}^{n \times n}$ with $\Sigma_{i,i} = 1, \forall i \in [n]$, which is the covariance matrix of a centered distribution over $\{-1, 1\}^n$, i.e. $\mathbb{E}_\mu[\mathbf{x}_i \mathbf{x}_j] = \Sigma_{i,j}, \mathbb{E}_\mu[\mathbf{x}_i] = 0$, for a distribution $\mu : \{-1, 1\}^n \to \mathbb{R}$. We wish to produce a distribution $\tilde{\mu} : \{-1, 1\}^n \to \mathbb{R}$ with pairwise covariances that match the given ones up to constant factors, and entropy within a constant factor of the maximum entropy distribution with covariance $\Sigma$.
[6]

Before stating the result formally, it will be useful to define the following constant:

**Definition 3.1.** *Define the constant* $\mathcal{G} = \min_{t \in [-1,1]} \left\{ \frac{2}{\pi} \arcsin(t)/t \right\} \approx 0.64.$

We will prove the following main theorem:

**Theorem 3.1** (Main, approximate entropy principle). *For any positive-semidefinite matrix $\Sigma$ with $\Sigma_{i,i} = 1, \forall i$, there is an efficiently sampleable distribution $\tilde{\mu} : \{-1, 1\}^n \to \mathbb{R}$, which can be sampled as sign(g), where $g \sim \mathcal{N}(0, \Sigma + \beta I)$, and satisfies $\frac{\mathcal{G}}{1+\beta} \Sigma_{i,j} \leq E_{\tilde{\mu}}[\mathbf{x}_i \mathbf{x}_j] \leq \frac{1}{1+\beta} \Sigma_{i,j}$ and has entropy $H(\tilde{\mu}) \geq \frac{n}{25} \frac{(3^{1/4}\sqrt{\beta}-1)^2}{\sqrt{3}\beta}$, where $\beta \geq \frac{1}{3^{1/2}}$.*

Note $\tilde{\mu}$ is in fact very close to the the one which is classically used to round semidefinite relaxations for solving the MAX-CUT problem. [GW95] We will prove Theorem 3.1 in two parts – by first lower bounding the entropy of $\tilde{\mu}$, and then by bounding the moments of $\tilde{\mu}$.

**Theorem 3.2.** *The entropy of the distribution $\tilde{\mu}$ satisfies $H(\tilde{\mu}) \geq \frac{n}{25} \frac{(3^{1/4}\sqrt{\beta}-1)^2}{\sqrt{3}\beta}$ when $\beta \geq \frac{1}{3^{1/2}}$.*

*Proof.* A sample $g$ from $\mathcal{N}(0, \tilde{\Sigma})$ can be produced by sampling $g_1 \sim \mathcal{N}(0, \Sigma)$, $g_2 \sim \mathcal{N}(0, \beta I)$ and setting $g = g_1 + g_2$. The sum of two multivariate normals is again a multivariate normal. Furthermore, the mean of $g$ is 0, and since $g_1, g_2$ are independent, the covariance of $g$ is $\Sigma + \beta I = \tilde{\Sigma}$.

Let's denote the random variable $\mathbb{Y} = \text{sign}(g_1 + g_2)$ which is distributed according to $\tilde{\mu}$. We wish to lower bound the entropy of $\mathbb{Y}$. Toward that goal, denote the random variable $\mathbb{S} := \{i \in [n] : |(g_1)_i| \leq cD\}$ for $c, D$ to be chosen. Then, we have: for $\gamma = \frac{c-1}{c}$,

$$H(\mathbb{Y}) \geq H(\mathbb{Y}|\mathbb{S}) = \sum_{S \subseteq [n]} \Pr[\mathbb{S} = S] H(\mathbb{Y}|\mathbb{S} = S) \geq \sum_{S \subseteq [n], |S| \geq \gamma n} \Pr[\mathbb{S} = S] H(\mathbb{Y}|\mathbb{S} = S)$$

where the first inequality follows since conditioning doesn't decrease entropy, and the latter by the non-negativity of entropy. Continue the calculation we can get:

$$\sum_{S \subseteq [n], |S| \geq \gamma n} \Pr[\mathbb{S} = S] H(\mathbb{Y}|\mathbb{S} = S) \geq \sum_{S \subseteq [n], |S| \geq \gamma n} \Pr[\mathbb{S} = S] \min_{S \subseteq [n], |S| \geq \gamma n} H(\mathbb{Y}|\mathbb{S} = S)$$

$$= \Pr[|\mathbb{S}| \geq \gamma n] \min_{S \subseteq [n], |S| \geq \gamma n} H(\mathbb{Y}|\mathbb{S} = S)$$

We will lower bound $\Pr[|\mathbb{S}| \geq \gamma n]$ first. Notice that $\mathbb{E}[\sum_{i=1}^n (g_1)_i^2] = n$, therefore by Markov's inequality, $\Pr\left[\sum_{i=1}^n (g_1)_i^2 \geq Dn\right] \leq \frac{1}{D}$. On the other hand, if $\sum_{i=1}^n (g_1)_i^2 \leq Dn$, then $|\{i : (g_1)_i^2 \geq cD\}| \leq \frac{n}{c}$, which means that $|\{i : (g_1)_i^2 \leq cD\}| \geq n - \frac{n}{c} = \frac{(c-1)n}{c} = \gamma n$. Putting things together, this means $\Pr[|\mathbb{S}| \geq \gamma n] \geq 1 - \frac{1}{D}$.

It remains to lower bound $\min_{S \subseteq [n], |S| \geq \gamma n} H(\mathbb{Y}|\mathbb{S} = S)$. For every $S \subseteq [n], |S| \geq \gamma n$, denote by $\mathbb{Y}_S$ the coordinates of $\mathbb{Y}$ restricted to $S$, we get

$$H(\mathbb{Y}|\mathbb{S} = S) \geq H(\mathbb{Y}_S|\mathbb{S} = S) \geq H_\infty(\mathbb{Y}_S|\mathbb{S} = S) = -\log(\max_{y_S} \Pr[\mathbb{Y}_S = y_S|\mathbb{S} = S])$$

(where $H_\infty$ is the min-entropy) so we only need to bound $\max_{y_S} \Pr[\mathbb{Y}_S = y_S | \mathbb{S} = S]$

We will now, for any $y_S$, upper bound $\Pr[\mathbb{Y}_S = y_S | \mathbb{S} = S]$. Recall that the event $\mathbb{S} = S$ implies that $\forall i \in S, |(g_1)_i| \le cD$. Since $g_2$ is independent of $g_1$, we know that for every fixed $g \in \mathbb{R}^n$:

$$\Pr[\mathbb{Y}_S = y_S | \mathbb{S} = S, g_1 = g] = \Pi_{i \in S} \Pr[\text{sign}([g]_i + [g_2]_i) = y_i]$$

For a fixed $i \in [S]$, consider the term $\Pr[\text{sign}([g]_i + [g_2]_i) = y_i]$. Without loss of generality, let's assume $[g]_i > 0$ (the proof is completely symmetric in the other case). Then, since $[g]_i$ is positive and $g_2$ has mean 0, we have $\Pr[[g]_i + (g_2)_i < 0] \le \frac{1}{2}$.

Moreover,

$$
\begin{aligned}
\Pr\left[[g]_i + [g_2]_i > 0\right] &= \Pr[[g_2]_i > 0] \Pr\left[[g]_i + [g_2]_i > 0 \mid [g_2]_i > 0\right] \\
&\quad + \Pr[[g_2]_i < 0] \Pr\left[[g]_i + [g_2]_i > 0 \mid [g_2]_i < 0\right]
\end{aligned}
$$

The first term is upper bounded by $\frac{1}{2}$ since $\Pr[[g_2]_i > 0] \le \frac{1}{2}$. The second term we will bound using standard Gaussian tail bounds:

$$
\begin{aligned}
\Pr\left[[g]_i + [g_2]_i > 0 \mid [g_2]_i < 0\right] &\le \Pr\left[|[g_2]_i| \le |[g]_i| \mid [g_2]_i < 0\right] \\
&= \Pr[|[g_2]_i| \le |[g]_i|] \le \Pr[([g_2]_i)^2 \le cD] \\
&= 1 - \Pr[([g_2]_i)^2 > cD] \\
&\le 1 - \frac{2}{\sqrt{2\pi}} \exp\left(-cD/2\beta\right) \left(\sqrt{\frac{\beta}{cD}} - \left(\sqrt{\frac{\beta}{cD}}\right)^3\right)
\end{aligned}
$$

which implies

$$\Pr[[g_2]_i < 0] \Pr[[g]_i + [g_2]_i > 0 \mid [g_2]_i < 0] \le \frac{1}{2}\left(1 - \frac{2}{\sqrt{2\pi}} \exp\left(-cD/2\beta\right) \left(\sqrt{\frac{\beta}{cD}} - \left(\sqrt{\frac{\beta}{cD}}\right)^3\right)\right)$$

Putting together, we have

$$\Pr[\text{sign}((g_1)_i + (g_2)_i) = y_i] \le 1 - \frac{1}{\sqrt{2\pi}} \exp\left(-cD/2\beta\right) \left(\sqrt{\frac{\beta}{cD}} - \left(\sqrt{\frac{\beta}{cD}}\right)^3\right)$$

Together with the fact that $|\mathbb{S}| \ge \gamma n$ we get

$$\Pr[\mathbb{Y}_S = y_S | \mathbb{S} = s, g_1 = g] \le \left[1 - \frac{1}{\sqrt{2\pi}} \exp\left(-cD/2\beta\right) \left(\sqrt{\frac{\beta}{cD}} - \left(\sqrt{\frac{\beta}{cD}}\right)^3\right)\right]^{\gamma n}$$

which implies that

$$H(\mathbb{Y}) \ge -\left(1 - \frac{1}{D}\right) \frac{(c-1)n}{c} \log\left[1 - \frac{1}{\sqrt{2\pi}} \exp\left(-cD/2\beta\right) \left(\sqrt{\frac{\beta}{cD}} - \left(\sqrt{\frac{\beta}{cD}}\right)^3\right)\right]$$

By setting $c = D = 3^{1/4}\sqrt{\beta}$ and a straightforward (albeit unpleasant) calculation, we can check that $H(\mathbb{Y}) \ge \frac{n}{25} \frac{(3^{1/4}\sqrt{\beta}-1)^2}{\sqrt{3\beta}}$, as we need.

$\square$

We next show that the moments of the distribution are preserved up to a constant $\frac{\mathcal{G}}{1+\beta}$.

**Lemma 3.1.** *The distribution $\tilde{\mu}$ has $\frac{\mathcal{G}}{1+\beta}\Sigma_{i,j} \le E_{\tilde{\mu}}[X_i X_j] \le \frac{1}{1+\beta}\Sigma_{i,j}$*

*Proof.* Consider the Gram decomposition of $\tilde{\Sigma}_{i,j} = \langle v_i, v_j \rangle$. Then, $\mathcal{N}(0, \tilde{\Sigma})$ is in distribution equal to $(\text{sign}(\langle v_1, s \rangle), \ldots, \text{sign}(\langle v_n, s \rangle))$ where $s \sim \mathcal{N}(0, I)$. Similarly as in the analysis of Goemans-Williamson [GW95], if $\bar{v}_i = \frac{1}{\|v_i\|} v_i$, we have $\mathcal{G}\langle \bar{v}_i, \bar{v}_j \rangle \leq E_{\tilde{\mu}}[X_i X_j] = \frac{2}{\pi} \arcsin(\langle \bar{v}_i, \bar{v}_j \rangle) \leq \langle \bar{v}_i, \bar{v}_j \rangle$. However, since $\langle \bar{v}_i, \bar{v}_j \rangle = \frac{1}{\|v_i\| \|v_j\|} \langle v_i, v_j \rangle = \frac{1}{\|v_i\| \|v_j\|} \tilde{\Sigma}_{i,j} = \frac{1}{\|v_i\| \|v_j\|} \Sigma_{i,j}$ and $\|v_i\| = \sqrt{\tilde{\Sigma}_{i,i}} = \sqrt{1 + \beta}, \forall i \in [1, n]$, we get that $\frac{\mathcal{G}}{1 + \beta} \Sigma_{i,j} \leq E_{\tilde{\mu}}[X_i X_j] \leq \frac{1}{1 + \beta} \Sigma_{i,j}$ as we want. $\qquad\square$

Lemma 3.2 and 3.1 together imply Theorem 3.1.

## 4   Provable bounds for variational methods

We will in this section consider applications of the approximate maximum entropy principles we developed for calculating partition functions of Ising models. Before we dive into the results, we give brief preliminaries on variational methods and pseudo-moment convex relaxations.

**Preliminaries on variational methods and pseudo-moment convex relaxations** Recall, variational methods are based on the following simple lemma, which characterizes $\log \mathcal{Z}$ as the solution of an optimization problem. It essentially dates back to Gibbs [Ell12], who used it in the context of statistical mechanics, though it has been rediscovered by machine learning researchers [WJ08]:

**Lemma 4.1** (Variational characterization of $\log \mathcal{Z}$). *Let us denote by $\mathcal{M}$ the polytope of distributions over $\{-1, 1\}^n$. Then,*

$$\log \mathcal{Z} = \max_{\mu \in \mathcal{M}} \left\{ \sum_t J_t \mathbb{E}_\mu[\phi_t(\mathbf{x})] + H(\mu) \right\} \tag{1}$$

While the above lemma reduces calculating $\log \mathcal{Z}$ to an optimization problem, optimizing over the polytope $\mathcal{M}$ is impossible in polynomial time. We will proceed in a way which is natural for optimization problems – by instead optimizing over a relaxation $\mathcal{M}'$ of that polytope.

The relaxation will be associated with the degree-2 Lasserre hierarchy. Intuitively, $\mathcal{M}'$ has as variables tentative pairwise moments of a distribution of $\{-1, 1\}^n$, and it imposes all constraints on the moments that hold for distributions over $\{-1, 1\}^n$. To define $\mathcal{M}'$ more precisely we will need the following notion: (for a more in-depth review of moment-based convex hierarchies, the reader can consult [BKS14])

**Definition 4.1.** *A degree-2 pseudo-moment* [7] $\tilde{\mathbb{E}}_\nu[\cdot]$ *is a linear operator mapping polynomials of degree 2 to $\mathbb{R}$, such that $\tilde{\mathbb{E}}_\nu[\mathbf{x}_i^2] = 1$, and $\tilde{\mathbb{E}}_\nu[p(\mathbf{x})^2] \geq 0$ for any polynomial $p(\mathbf{x})$ of degree 1.*

We will be optimizing over the polytope $\mathcal{M}'$ of all degree-2 pseudo-moments, i.e. we will consider solving

$$\max_{\tilde{\mathbb{E}}_\nu[\cdot] \in \mathcal{M}'} \left\{ \sum_t J_t \tilde{\mathbb{E}}_\nu[\phi_t(\mathbf{x})] + \tilde{H}(\tilde{\mathbb{E}}_\nu[\cdot]) \right\}$$

where $\tilde{H}$ will be a proxy for the entropy we will have to define (since entropy is a global property that depends on all moments, and $\tilde{\mathbb{E}}_\nu$ only contains information about second order moments).

To see this optimization problem is convex, we show that it can easily be written as a semidefinite program. Namely, note that the pseudo-moment operators are linear, so it suffices to define them over monomials only. Hence, the variables will simply be $\tilde{\mathbb{E}}_\nu(\mathbf{x}_S)$ for all monomials $\mathbf{x}_S$ of degree at most 2. The constraints $\tilde{\mathbb{E}}_\nu[\mathbf{x}_i^2] = 1$ then are clearly linear, as is the "energy part" of the objective function. So we only need to worry about the constraint $\tilde{\mathbb{E}}_\nu[p(\mathbf{x})^2] \geq 0$ and the entropy functional.

We claim the constraint $\tilde{\mathbb{E}}_\nu[p(\mathbf{x})^2] \geq 0$ can be written as a PSD constraint: namely if we define the matrix $Q$, which is indexed by all the monomials of degree at most 1, and it satisfies $Q(\mathbf{x}_S, \mathbf{x}_T) = \tilde{\mathbb{E}}_\nu[\mathbf{x}_S \mathbf{x}_T]$. It is easy to see that $\tilde{\mathbb{E}}_\nu[p(\mathbf{x})^2] \geq 0 \equiv Q \succeq 0$.

Hence, the final concern is how to write an expression for the entropy in terms of the low-order moments, since entropy is a global property that depends on all moments. There are many candidates for this in machine learning are like Bethe/Kikuchi entropy, tree-reweighted Bethe entropy, log-determinant etc. However, in the worst case – none of them come with any guarantees. We will in fact show that the entropy functional is not an issue – we will relax the entropy trivially to $n$.

Given all of this, the final relaxation we will consider is:

$$\max_{\tilde{\mathbb{E}}_\nu[\cdot]\in\mathcal{M}'} \left\{ \sum_t J_t \tilde{\mathbb{E}}_\nu[\phi_t(\mathbf{x})] + n \right\} \tag{2}$$

From the prior setup it is clear that the solution to 2 is an upper bound to $\log \mathcal{Z}$. To prove a claim like Theorem 2.3 or Theorem 2.4, we will then provide a *rounding* of the solution. In this instance, this will mean producing a *distribution* $\tilde{\mu}$ which has the value of $\sum_t J_t \mathbb{E}_{\tilde{\mu}}[\phi_t(\mathbf{x})] + H(\tilde{\mu})$ comparable to the value of the solution. Note this is slightly different than the usual requirement in optimization, where one cares only about producing a single $\mathbf{x} \in \{-1, 1\}^n$ with comparable value to the solution. Our distribution $\tilde{\mu}$ will have entropy $\Omega(n)$, and preserves the "energy" portion of the objective $\sum_t J_t \mathbb{E}_\mu[\phi_t(\mathbf{x})]$ up to a comparable factor to what is achievable in the optimization setting.

**Warmup: exponential family analogue of MAX-CUT** As a warmup, to illustrate the basic ideas behind the above rounding strategy, before we consider Ising models we consider the exponential family analogue of MAX-CUT. It is defined by the functionals $\phi_{i,j}(\mathbf{x}) = (\mathbf{x}_i - \mathbf{x}_j)^2$. Concretely, we wish to approximate the partition function of the distribution $\mu(\mathbf{x}) \propto \exp\left( \sum_{i,j} J_{i,j}(\mathbf{x}_i - \mathbf{x}_j)^2 \right)$.

We will prove the following simple observation:

**Observation 4.1.** *The relaxation 2 provides a factor 2 approximation of* $\log \mathcal{Z}$.

*Proof.* We proceed as outlined in the previous section, by providing a rounding of 2. We point out again, unlike the standard case in optimization, where typically one needs to produce an assignment of the variables, because of the entropy term here it is crucial that the rounding produces a *distribution*.

The distribution $\tilde{\mu}$ we produce here will be especially simple: we will round each $x_i$ independently with probability $\frac{1}{2}$. Then, clearly $H(\tilde{\mu}) = n$. On the other hand, we similarly have $\Pr_{\tilde{\mu}}[(x_i - x_j)^2 = 1] = \frac{1}{2}$, since $x_i$ and $x_j$ are rounded independently. Hence, $E_{\tilde{\mu}}[(x_i - x_j)^2] \geq \frac{1}{2}$. Altogether, this implies $\sum_{i,j} J_{i,j} E_{\tilde{\mu}}[(x_i - x_j)^2] + H(\tilde{\mu}) \geq \frac{1}{2}\left( \sum_{i,j} J_{i,j} E_\nu[(x_i - x_j)^2] + n \right)$ as we needed.

□

## 4.1 Ising models

We proceed with the main results of this section on Ising models, which is the case where $\phi_{i,j}(\mathbf{x}) = \mathbf{x}_i\mathbf{x}_j$. We will split into the ferromagnetic and general case separately, as outlined in Section 2.

To be concrete, we will be given potentials $J_{i,j}$, and we wish to calculate the partition function of the Ising model $\mu(\mathbf{x}) \propto \exp(\sum_{i,j} J_{i,j}\mathbf{x}_i\mathbf{x}_j)$.

**Ferromagnetic case**

Recall, in the ferromagnetic case of Ising model, we have the conditions that the potentials $J_{i,j} > 0$. We will provide a convex relaxation which has a constant factor approximation in this case. First, recall the famous first Griffiths inequality due to Griffiths [Gri67] which states that in the ferromagnetic case, $\mathbb{E}_\mu[\mathbf{x}_i\mathbf{x}_j] \geq 0, \forall i, j$.

Using this inequality, we will look at the following natural strenghtening of the relaxation 2:

$$\max_{\tilde{\mathbb{E}}_\nu[\cdot]\in\mathcal{M}';\tilde{\mathbb{E}}_\nu[\mathbf{x}_i\mathbf{x}_j]\geq 0,\forall i,j} \left\{ \sum_t J_t \tilde{\mathbb{E}}_\nu[\phi_t(\mathbf{x})] + n \right\} \tag{3}$$

We will prove the following theorem, as a straightforward implication of our claims from Section 3:

**Theorem 4.1.** *The relaxation 3 provides a factor 50 approximation of* $\log \mathcal{Z}$.

*Proof.* Notice, due to Griffiths' inequality, 3 is in fact a relaxation of the Gibbs variational principle and hence an upper bound)of $\log \mathcal{Z}$. Same as before, we will provide a rounding of 3. We will use the distribution $\tilde{\mu}$ we designed in Section 3 the sign of a Gaussian with covariance matrix $\Sigma + \beta I$, for a $\beta$ which we will specify. By Lemma 3.2, we then have $H(\tilde{\mu}) \geq \frac{n}{25} \frac{(3^{1/4}\sqrt{\beta}-1)^2}{\sqrt{3}\beta}$ whenever $\beta \geq \frac{1}{3^{1/2}}$.

By Lemma 3.1, on the other hand, we can prove that $E_{\tilde{\mu}}[\mathbf{x}_i \mathbf{x}_j] \geq \dfrac{\mathcal{G}}{1+\beta} \tilde{\mathbb{E}}_\nu[\mathbf{x}_i \mathbf{x}_j]$

By setting $\beta = 21.8202$, we get $\frac{n}{25} \frac{(3^{1/4}\sqrt{\beta}-1)^2}{\sqrt{3}\beta} \geq 0.02$ and $\frac{\mathcal{G}}{1+\beta} \geq 0.02$, which implies that

$$\sum_{i,j} J_{i,j} \mathbb{E}_{\tilde{\mu}}[\mathbf{x}_i \mathbf{x}_j] + H(\tilde{\mu}) \geq 0.02 \left( \sum_{i,j} J_{i,j} \tilde{\mathbb{E}}_\nu[\mathbf{x}_i \mathbf{x}_j] + n \right)$$

which is what we need. $\qquad\square$

Note that the above proof does not work in the general Ising model case: when $\tilde{\mathbb{E}}_\nu[\mathbf{x}_i \mathbf{x}_j]$ can be either positive or negative, even if we preserved each $\tilde{\mathbb{E}}_\nu[\mathbf{x}_i \mathbf{x}_j]$ up to a constant factor, this may not preserve the sum $\sum_{i,j} J_{i,j} \tilde{\mathbb{E}}_\nu[\mathbf{x}_i \mathbf{x}_j]$ due to cancellations in that expression.

**General Ising models case**

Finally, we will tackle the general Ising model case. As noted in the previous section, the straightforward application of the results proven in Section 3 doesn't work, so we have to consider a different rounding – again inspired by roundings used in optimization.

The intuition is the same as in the ferromagnetic case: we wish to design a rounding which preserves the "energy" portion of the objective, while having a high entropy. In the previous section, this was achieved by modifying the Goemans-Williamson rounding so that it produces a high-entropy distribution. We will do a similar thing here, by modifying rounding due to [CW04] and [AMMN06].

The convex relaxation we will consider will just be the basic one: 2 and we will prove the following two theorems:

**Theorem 4.2.** *The relaxation 2 provides a factor $O(\log n)$ approximation to $\log \mathcal{Z}$ when $\phi_{i,j}(\mathbf{x}) = \mathbf{x}_i \mathbf{x}_j$.*

**Theorem 4.3.** *The relaxation 2 provides a factor $O(\log(\chi(G)))$ approximation to $\log \mathcal{Z}$ when $\phi_{i,j}(\mathbf{x}) = \mathbf{x}_i \mathbf{x}_j$ for $i, j \in E(G)$ of some graph $G = (V(G), E(G))$, and $\chi(G)$ is the chromatic number of $G$.*

Since the chromatic number of a graph is bounded by $n$, the second theorem is in fact strictly stronger than the first, however the proof of the first theorem uses less heavy machinery, and is illuminating enough to be presented on its own.

Due to space constraints, the proofs of these theorems are forwarded to the appendix.

## 5   Conclusion

In summary, we presented computationally efficient approximate versions of the classical max-entropy principle by [Jay57]: efficiently sampleable distributions which preserve given pairwise moments up to a multiplicative constant factor, while having entropy within a constant factor of the maximum entropy distribution matching those moments. Additionally, we applied our insights to designing provable variational methods for Ising models which provide comparable guarantees for approximating the log-partition function to those in the optimization setting. Our methods are based on convex relaxations of the standard variational principle due to Gibbs, and are extremely generic and we hope they will find applications for other exponential families.

## Footnotes

[1] There is a more general way to state this principle over an arbitrary domain, not just the hypercube, but for clarity in this paper we will focus on the hypercube only.

[2]In fact, we produce a distribution with entropy $\Omega(n)$, which implies the latter claim since the maximum entropy of any distribution of over $\{-1, 1\}^n$ is at most $n$

[3]Theorem 2.4 is strictly more general than Theorem 2.3, however the proof of Theorem 2.3 uses less heavy machinery and is illuminating enough that we feel merits being presented as a separate theorem.

[4]In some sense, it is possible to give provable bounds for Bethe-entropy based relaxations, via analyzing belief propagation directly, which has been done in cases where there is correlation decay and the graph is locally tree-like. [WJ08] has a detailed overview of such results.

[5]More precisely, they prove that in the case when $\forall i, j, \Delta|J_{i,j}| \leq \frac{\Delta}{n^2} \sum_{i,j} |J_{i,j}|$, one can get an additive $\epsilon(\sum_{i,j} J_{i,j})$ approximation to $\log \mathcal{Z}$ in time $n^{O(\frac{\Delta}{\epsilon^2})}$.

[6]Note for a distribution over $\{-1, 1\}^n$, the maximal entropy a distribution can have is $n$, which is achieved by the uniform distribution.

[7]The reason $\tilde{\mathbb{E}}_\nu[\cdot]$ is called a pseudo-moment, is that it behaves like the moments of a distribution $\nu : \{-1, 1\}^n \to [0, 1]$, albeit only over polynomials of degree at most 2.

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
