[Supplementary Material · entropy_main_nomaxcut_long.pdf]

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

Before delving into the proof of Theorem 4.2, we review the rounding used by [CW04] in the case of maximizing quadratic forms:

---

**Algorithm 1** Quadratic form rounding by [CW04]

---

1: Input: A pseudo-moment matrix $\Sigma_{i,j} = \mathbb{E}_\nu[\mathbf{x}_i\mathbf{x}_j]$
2: Output: A sample $\mathbf{x}$ from a distribution $\rho$
3: Sample $g$ from the standard Gaussian $N(0, I)$.
4: Consider the vector $h$, such that $h_i = g_i/T, T = \sqrt{4\log n}$
5: Consider the vector $r$, such that $r_i = \frac{h_i}{|h_i|}$, if $|h_i| > 1$, and $r_i = h_i$ otherwise.
6: Produce the rounded vector $\mathbf{x} \in \{-1, 1\}^n$, s.t.

$$\mathbf{x}_i = \left\{ \begin{array}{ll} +1, & \text{with probability } \frac{1+r_i}{2} \\ -1, & \text{with probability } \frac{1-r_i}{2} \end{array} \right\}$$

---

---

**Algorithm 2** Scaled down quadratic form rounding

---

1: Input: A pseudo-moment matrix $\Sigma_{i,j} = \mathbb{E}_\nu[\mathbf{x}_i\mathbf{x}_j]$
2: Output: A sample $\mathbf{x}$ from a distribution $\tilde{\mu}$
3: Sample $g$ from the standard Gaussian $N(0, I)$.
4: Consider the vector $h$, such that $h_i = g_i/T, T = \sqrt{4\log n}$
5: Consider the vector $r$, such that $r'_i = \frac{1}{2}\frac{h_i}{|h_i|}$, if $|h_i| > 1$, and $r'_i = \frac{1}{2}h_i$ otherwise.
6: Produce the rounded vector $\mathbf{x} \in \{-1, 1\}^n$, s.t.

$$\mathbf{x}_i = \left\{ \begin{array}{ll} +1, & \text{with probability } \frac{1+r_i}{2} \\ -1, & \text{with probability } \frac{1-r_i}{2} \end{array} \right\}$$

---

With that in hand, we can prove Theorem 4.2

*Proof of Theorem 4.2.* The proof again consists of exhibiting a rounding. Our rounding will essentially be the same as [CW04], except in step 3, we will produce a vector $r'_i$ by scaling down the vector $r_i$ by 2 coordinate-wise. For full clarity, the rounding is presented in Algorithm 2.

We again, need to analyze the entropy and the moments of the distribution $\tilde{\mu}$ that this rounding produces. Let us focus on the entropy first.

Since conditioning does not decrease entropy, it's true that $H(\tilde{\mu}) = H(\mathbf{x}) \geq H(\mathbf{x}|r)$, so it suffices to lower bound that quantity. However, note that it holds that $r_i \leq \frac{1}{2}$, and each $x_i$ is rounded independently conditional on $r_i$, so we have:

$$H(\mathbf{x}|r) = \sum_i H(\mathbf{x}_i|r_i) = \sum_i \left( \frac{1+r_i}{2} \log\left(\frac{1+r_i}{2}\right) + \frac{1-r_i}{2}\left(\frac{1-r_i}{2}\right) \right) \geq \left(2 - \frac{3}{4}\log 3\right) n$$

Consider now the moments of the distribution.

Let us denote the distribution that the rounding 1 produces by $\rho$. By Theorem 1 in [CW04], we have

$$\sum_{i,j} J_{i,j}\mathbb{E}_\rho[\mathbf{x}_i\mathbf{x}_j] \geq O\left(\frac{1}{\log n}\right)\sum_{i,j} J_{i,j}\mathbb{E}_\nu[\mathbf{x}_i\mathbf{x}_j]$$

Additional, both our and the [CW04] roundings are such that $\mathbb{E}_\rho[\mathbf{x}_i\mathbf{x}_j] = \mathbb{E}_{r_i}\mathbb{E}_{\mathbf{x}|r_i}[\mathbf{x}_i\mathbf{x}_j]$ and $\mathbb{E}_{\tilde{\mu}}[\mathbf{x}_i\mathbf{x}_j] = \mathbb{E}_{r'_i}\mathbb{E}_{\mathbf{x}|r'_i}[\mathbf{x}_i\mathbf{x}_j]$. Furthermore, as noted in [CW04], it is easy to check that $\mathbb{E}[\mathbf{x}_i\mathbf{x}_j|r'_i, r'_j] = r'_i r'_j$ and obviously $r'_i = 2r_i$ in distribution, so we have:

$$\mathbb{E}_{\tilde{\mu}}[\mathbf{x}_i\mathbf{x}_j] = \mathbb{E}_{r'_i}\mathbb{E}_{\mathbf{x}|r'_i}[\mathbf{x}_i\mathbf{x}_j] = \frac{1}{4}\mathbb{E}_{r_i}\mathbb{E}_{\mathbf{x}|r_i}[\mathbf{x}_i\mathbf{x}_j] = \frac{1}{4}\mathbb{E}_\rho[\mathbf{x}_i\mathbf{x}_j]$$

But, this directly implies

$$\sum_{i,j} J_{i,j}\mathbb{E}_{\tilde{\mu}}[\mathbf{x}_i\mathbf{x}_j] = \frac{1}{4}\sum_{i,j} J_{i,j}\mathbb{E}_\rho[\mathbf{x}_i\mathbf{x}_j] \geq O\left(\frac{1}{\log n}\right)\sum_{i,j} J_{i,j}\mathbb{E}_\nu[\mathbf{x}_i\mathbf{x}_j]$$

as we needed. $\square$

Next, we prove the more general Theorem 4.3.

Before proceeding, let's recall for completeness the following definition of a chromatic number.

**Definition 4.2** (Chromatic number). *The chromatic number $\chi(G)$ of a graph $G = (V(G), E(G))$ is defined as the minimum number of colors in a coloring of the vertices $V(G)$, such that vertices $i, j : (i, j) \in E(G)$ are colored with the same color.*

Also, let us denote by $\mathcal{S}^{n-1}$ the set of unit vectors in $\mathbb{R}^n$ and $L_\infty[0, 1]$ the set of (essentially) bounded functions: the functions which are bounded except on a set of measure zero.

Then, we can recall Theorem 3.3 from [AMMN06]:

**Theorem 4.4** ([AMMN06]). *There exists an absolute constant $c$ such that the following holds: Let $G = (V(G), E(G))$ be an undirected graph on $n$ vertices without self-loops[8], let $\chi(G)$ be the chromatic number of $G$. Then for every function $f : V(G) \to \mathcal{S}^{n-1}$, there exists a function $F : V \to L_\infty[0, 1]$ so that for every $i \in V(G)$, $\|F(i)\|_\infty \leq \sqrt{c\chi(G)}$ and for every $(i, j) \in E(G)$,*

$$\langle f(i), f(j) \rangle = \int_0^1 F(i)(t) F(j)(t) dt$$

Now, we can prove Theorem 4.3

*Proof of Theorem 4.3.* The proof is similar, though a little more complicated than the proof of Theorem 4.2.

Let $\tilde{\mathbb{E}}_\nu[\cdot]$ be the solution of the relaxation. By matrix formulation of the pseudo-moment relaxation in Section 4, we know that $\tilde{\mathbb{E}}_\nu[\mathbf{x}_i \mathbf{x}_j] = \langle f(i), f(j) \rangle$ for some unit vectors $f(i), f(j)$.

Hence, by theorem 4.4, there exists a function $F : V \to L_\infty[0, 1]$ so that for every $i \in V(G)$, $\|F(i)\|_\infty \leq \sqrt{c\chi(G)}$ and for every $(i, j) \in E(G)$,

$$\tilde{\mathbb{E}}_\nu[\mathbf{x}_i \mathbf{x}_j] = \int_0^1 F(i)(t) F(j)(t) dt$$

Consider the following rounding:

- Pick a $t$ uniformly at random from $[0, 1]$.

- Consider the function $h_t : V \to R$, such that $h_t(i) = \frac{F(i)(t)}{2\sqrt{c\chi(G)}}$

- Produce the rounded vector $\mathbf{x} \in \{-1, 1\}^{V(G)}$, s.t.

$$\mathbf{x}_i = \left\{ \begin{array}{ll} +1, & \text{with probability } \frac{1 + h_t(i)}{2} \\ -1, & \text{with probability } \frac{1 - h_t(i)}{2} \end{array} \right\}$$

Note importantly that the algorithm does not need to perform this rounding – it is for the analysis of the approximation factor of the relaxation. Therefore, we need not construct it algorithmically.

Let us denote this distribution as $\tilde{\mu}$. We first show that $\tilde{\mu}$ has entropy at least $\left(2 - \frac{3}{4} \log 3\right) n$. Note that each $\mathbf{x}_i$ are round independently conditional on $t$. Moreover, since $\|F(v)\|_\infty \leq \sqrt{c\chi(G)}$, we know that $h_t(v) \leq \frac{1}{2}$. Therefore, for every fixed $t_0 \in [0, 1]$

$$\begin{aligned} H(\tilde{\mu} \mid t = t_0) &= \sum_{i \in V(G)} H(\mathbf{x}_i \mid t = t_0) \\ &= \sum_{i \in V(G)} \left( \frac{1 + h_{t_0}(v)}{2} \log \frac{1 + h_{t_0}(v)}{2} + \frac{1 - h_{t_0}(v)}{2} \log \frac{1 - h_{t_0}(v)}{2} \right) \\ &\geq \left( 2 - \frac{3}{4} \log 3 \right) n \end{aligned}$$

Integrating over $t_0$ we get that $H(\tilde{\mu}) \geq \left(2 - \frac{3}{4}\log 3\right) n$.

Next, we will show that $\tilde{\mu}$ preserves the "energy" part of the objective up to a multiplicative factor $O(\log \chi(G))$: Consider each edge $(i,j) \in E(G)$. We have:

$$\mathbb{E}_{\tilde{\mu}}[\mathbf{x}_i \mathbf{x}_j] =$$

$$\int_0^1 \left( \frac{[1+h_t(i)][1+h_t(j)]}{4} + \frac{[1-h_t(i)][1-h_t(j)]}{4} - \frac{[1+h_t(i)][1-h_t(j)]}{4} - \frac{[1-h_t(i)][1+h_t(j)]}{4} \right) dt$$

$$= \int_0^1 h_t(i)h_t(j)dt = \frac{1}{4c\chi(G)} \int_0^1 F(i)(t)F(j)(t)dt = \frac{1}{4c\chi(G)} \tilde{\mathbb{E}}_\nu[\mathbf{x}_i \mathbf{x}_j]$$

This implies that

$$\sum_{i,j \in E(G)} J_{i,j} \mathbb{E}_{\tilde{\mu}}[\mathbf{x}_i \mathbf{x}_j] \geq \frac{1}{4c\chi(G)} \sum_{i,j \in E(G)} J_{i,j} \tilde{\mathbb{E}}_\nu[\mathbf{x}_i \mathbf{x}_j]$$

Therefore, the relaxation provides a factor $O(\chi(G))$ approximation of $\log \mathcal{Z}$, as we wanted.

$\square$

## 5  Conclusion

In summary, we presented computationally efficient approximate versions of the classical max-entropy principle by [Jay57]: efficiently sampleable distributions which preserve given pairwise moments up to a multiplicative constant factor, while having entropy within a constant factor of the maximum entropy distribution matching those moments. Additionally, we applied our insights to designing provable variational methods for Ising models which provide comparable guarantees for approximating the log-partition function to those in the optimization setting. Our methods are based on convex relaxations of the standard variational principle due to Gibbs, and are extremely generic and we hope they will find applications for other exponential families.

## Footnotes

[1]There is a more general way to state this principle over an arbitrary domain, not just the hypercube, but for clarity in this paper we will focus on the hypercube only.

[2]In fact, we produce a distribution with entropy $\Omega(n)$, which implies the latter claim since the maximum entropy of any distribution of over $\{-1,1\}^n$ is at most $n$

[3]Theorem 2.4 is strictly more general than Theorem 2.3, however the proof of Theorem 2.3 uses less heavy machinery and is illuminating enough that we feel merits being presented as a separate theorem.

[4]In some sense, it is possible to give provable bounds for Bethe-entropy based relaxations, via analyzing belief propagation directly, which has been done in cases where there is correlation decay and the graph is locally tree-like. [WJ08] has a detailed overview of such results.

[5]More precisely, they prove that in the case when $\forall i, j, \Delta|J_{i,j}| \leq \frac{\Delta}{n^2}\sum_{i,j}|J_{i,j}|$, one can get an additive $\epsilon(\sum_{i,j} J_{i,j})$ approximation to $\log \mathcal{Z}$ in time $n^{O(\frac{\Delta}{\epsilon^2})}$.

[6]Note for a distribution over $\{-1, 1\}^n$, the maximal entropy a distribution can have is $n$, which is achieved by the uniform distribution.

[7] The reason $\tilde{\mathbb{E}}_\nu[\cdot]$ is called a pseudo-moment, is that it behaves like the moments of a distribution $\nu : \{-1, 1\}^n \to [0, 1]$, albeit only over polynomials of degree at most 2.

[8]Meaning no edge connects a vertex with itself