[Reviews · NeurIPS 2016]

Reviewer 1

Summary

The authors provide efficiently sample-able approximations to max-entropy distributions given pairwise moments. They also provide bounds on approximating the log-partition function given pairwise approximating distributions for the Ising model.

Qualitative Assessment

The technical contributions of this paper are well-explained and appear correct. My main problem with the paper is motivation. I was never convinced that the maximum-entropy principle was useful, so being able to approximate maxent distributions doesn't seem especially important. The second part of the results are more interesting. As far as I understand, there are few bounds on log-partition functions. I'm not very familiar with the state of the literature, but this seems like an interesting result that could lead to further useful theorems or methods. The authors mention that their results might extent to other exponential families. However, without any other examples, and no experiments, I think this paper belongs in COLT rather than NIPS. If the paper took more steps towards turning their result into concrete methodological improvements, it might be able to be turned into a NIPS paper. Typos: Line 212 - in what sense are \phi functionals? aren't they just functions? Line 215 and 216. In black-and-white, it's confusing that you refer to the numeral 2 and equation 2 in the same way in the same sentence. Line 304 - isolated footnote

Confidence in this Review

1-Less confident (might not have understood significant parts)


Reviewer 2

Summary

The authors provide an approximation algorithm to the log-partition function of Ising models. The algorithm uses approximation algorithm following the famous Goemans-Williamson MAXCUT algorithm.

Qualitative Assessment

This is a nice paper, a bit of an odd match for NIPS (there are no numerical experiments, and in spite of claims of genericity and applicability to general exponential families, I remain unconvinced). The methods are elegant, though I did find the presentation a bit lacking. I would have loved a high-level detail of the proof steps and proof intuition, with pointers to precise sub-proposition statements and corresponding proofs. Right now, it is easy to get lost in the details, and what appears to me as the key moments of the proof are skimmed over quickly. For instance, lemma 3.1 deserved to be expanded upon (even the long version is a bit quick on details here) - this is especially since the GW proof technique is so elegant, it's always nice to include (even if similar to the original proof). Similarly, it seems to me the main theorems are in fact theorems 2.2-2.4; not theorem 2.1 (which has a host of odd constants and a bound on entropy which is not clearly interesting at first glance*). Theorem 2.1's existence seems justified by thm 2.2-2.4 - would it make sense to introduce 2.2-4 first, then 2.1 afterwards as a proof technique? Similarly, should the proof for 2.2-2.4 have contained slightly more details? * is the bound interesting in itself? It does not seem to relate to the actual entropy of the max-entropy distribution, so it was not clear to me the result was impressive or not. minor: - Given the paper does not build further than degree-2 pseudomoment, it's not clear that using the language from that hierarchy helps understanding - I think a clear (equation driven) definition of the polytope used in that section would have made for easier read. - Many equations are referred to by number - as in, 'solutions to 2' (line 203, 237); it would preferrable to either use 'solution to equation 2' , or 'solution to (2)'. - In proof of 3.1, can \mathcal{N(0,\tilde \Sigma)} - a continuous vector - really be *equal* in distribution to (sign(v1,s),...), - a binary vector? I realize a lot of the ideas in the paper are related to sampling discrete variables with covariance matrix similar to that of a Gaussian, but at this particular point of the proof, it seems odd. - I don't have a strong intuition for the proofs of the paper - I will say, however, that I am surprised that the entropy terms (which are the very reason there is a difference between MAP and posterior sampling) can be bounded in such trivial ways (bound by the entropy of the uniform), and still obtain interesting results. - In proof of theorem 3.2, line 154, I am surely missing something very simple, but how do have P(g2\leq g)\leq P(g2^2 \leq cD)? - I didn't quite get proof of observation 4.4- we have E_{\tilde \mu}[(x_i-x_j)^2] \geq 1/2. How does The expectation over \nu appear in the following equation?

Confidence in this Review

2-Confident (read it all; understood it all reasonably well)


Reviewer 3

Summary

Considering Ising models (with no local fields), a very interesting twist on the standard variational inference approach is developed to show a convex relaxation and rounding that yields a multiplicative factor approximation to log Z. The factor is O(log chi(G)) where chi(G) is the chromatic number of the model graph. For fully attractive (ferromagnetic) models, this improves to a constant factor of 50.

Qualitative Assessment

The paper is clear and well written with helpful background and remarks [I have not checked the proofs in detail but the ideas are presented clearly and I trust that the details are correct]. The methods used and results obtained are very interesting. To my knowledge they are novel and may prove useful in other work. Note that a multiplicative bound on Z (hence an additive bound on log Z) would be much stronger and more useful, but as noted in the paper, this is likely impossible. It is not clear how useful these results will be in the near term in practice, but still they are theoretically important and provide good leads for future work. The authors focus on `symmetric' Ising models with no local fields/singleton potentials, i.e. x_i \in {-1,+1} and E(x_i)=0 for all i. It is worth noting that if an Ising model does have local fields, it can always be transformed into a larger model without any local fields by adding one extra variable and encoding the original singleton potentials as edge potentials to the added variable. The larger model has exactly twice the partition function of the original model, e.g. see Weller ICML 2016, Uprooting and Rerooting Graphical Models. Further observations on the following would be welcome: A longer comment on the differences between the analysis here and in Ris16 (so expand on lines 84-88 and 105-108). Are there any results/conjectures on what might be the best possible approximation bounds for these problems, so we can see how close these results are? Minor points: Section 2 is very helpful as a clear statement of the main results and prior related work. It would be good to tie it to the later text more neatly: Definition 3.1 of the constant G should come before it is first used in Theorem 2.1. Clarify where exactly Theorems 2.1-2.4 are proved. line 32: a -> an 33: from from -> than 35-38: very interesting, in 37: delete one 'there' 41: The -> A (there are other important reasons to compute Z) 45-46: Perhaps mention that we know that the Bethe partition function estimate lower bounds the true value (i.e. Z_B \leq Z) for ferromagnetic models, see Ruozzi NIPS 2012, The Bethe partition function of log-supermodular graphical models, and Weller and Jebara NIPS 2014, Clamping variables and approximate inference. 50: If the configuration with max score is not unique, then log Z -> the max * the multiplicity of the max 57: Perhaps mention that better results can be obtained for various restrictions such as bounded treewidth (Wainwright and Jordan 2004, Treewidth-based conditions for exactness of the Sherali-Adams and Lasserre relaxations) or if certain minors are not present (Weller UAI 2016, Characterizing Tightness of LP Relaxations by Forbidding Signed Minors). 104: Perhaps move the footnote 3 to the end of 103 to save space. 118: theoretically much more poorly studied - not sure if that's true 121: delete 'what' 238: add space between bound) and of. Delete Same. 253: Insert 'a' before rounding 254, 256, 258 should the 2 be (2) as in \eqref References In several places, caps should be fixed, e.g. Grothendieck -> {G}rothendieck. I believe [BB] should be [BB08] from NIPS 2008.

Confidence in this Review

2-Confident (read it all; understood it all reasonably well)


Reviewer 4

Summary

The paper presents an approach to approximate maximum entropy principles with applications to estimating mean parameters along with partition functions for Ising models. Along with a review of previous well-known maximum entropy and variational methods, they present an approximate distribution that is not entropy maximizing, and thus does not require hardness proofs nor assumptions on the potential functions. The majority of the paper is dedicated to the proof of Theorem 3.1, stating that an efficiently sampleable distribution exists for a PSD covariance matrix with a specific minimum entropy, and to the description of 3 additional theorems stating that a convex programming relaxation exists for estimating the log-partition function up to specific multiplicative factors.

Qualitative Assessment

While the paper has no fatal flaws, there appear to be many questions that a reader may have while understanding their work. The most obvious question is the usefulness in application of the ideas presented. While there results are somewhat interesting because of the reduction in assumptions, in almost all applications of the Ising model as presented those assumptions described by [WJ08] and by others are completely reasonable, and lead to theoretical results much stronger than those presented here. In the case of the Ferromagnetic Ising, the authors themselves refer to the work by [JS93] which gives a poly-time approximate MCMC algorithm. The main result describing the efficient sampling distribution is definitely the highlight of the work, but most of the paper is dedicated to its proof rather than its ramifications. The section on variational bounds for the other three results are also dedicated to the explanation and proofs of the theorems (the last is allocated to the supplement). Again, it is hard for a reader to understand the application of these results. How can approximating the log-parition function up to a multiplicative factor of 50 be useful? In the supplement they outline a particular novel rounding algorithm which allows for their convex programming claim, but there is no mention of this in the paper. Though the paper could also use some significant clarity in language and presentation, the most concerning issue for this reviewer is the lack of application analysis and experiments. They provide theorems describing convex programming relaxations to generally intractable problems and do not present experimental results demonstrating the applicability of their methods. The problem of variational methods is extremely well-studied, particularly with Ising models. To provide a novel relaxation without showing the application significantly reduces the potential impact on a reader, even if the theory may be well supported.

Confidence in this Review

2-Confident (read it all; understood it all reasonably well)


Reviewer 5

Summary

The paper tries to establish a practical adaptation of maximum entropy principle, since the principle itself is impractical from a computational perspective. Given a covariance matrix, it introduces a distribution such that 1) it is easy to draw sample from, 2) the pairwise covariance of which are within constant approximation of the elements of given covariance matrix, and 3) its entropy is lower-bounded by linear term (which basically means it's close to the maximum entropy). Moreover, using the results from approximate maximum entropy, the paper finds approximation to the partition function of a particular member of exponential family -- namely Ising models. These approximation guarantees are established with the means of degree-2 pseudo-moment relaxations of the standard variational principle.

Qualitative Assessment

The paper takes a fairly novel approach to establish approximate maximum entropy. The usage of this approximation in variational methods and specifically on Ising models, however, is not quite clear to be useful since it depends on the constants in the asymptotic notation which are not clear. The organization of the paper is solid; nevertheless, there are some parts and connections that are hard to follow. Concretely the connection between Approximate Maximum Entropy and calculating the partition function in Ising models is not well-explained from reviewer's perspective.

Confidence in this Review

1-Less confident (might not have understood significant parts)